# Acute respiratory distress syndrome after SARS-CoV-2 infection on young adult population: International observational federated study based on electronic health records through the 4CE consortium

Bertrand Moal[1]*, Arthur Orieux[2], Thomas Ferté[3], Antoine Neuraz[4], Gabriel A. Brat[5], Paul Avillach[5], Clara-Lea Bonzel[5], Tianxi Cai[5], Kelly Cho[6], Sébastien Cossin[7], Romain Griffier[8], David A. Hanauer[9], Christian Haverkamp[10], Yuk-Lam Ho[8], Chuan Hong[5], Meghan R. Hutch[11], Jeffrey G. Klann[12], Trang T. Le[5], Ne Hooi Will Loh[13], Yuan Luo[14], Adeline Makoudjou[15], Michele Morris[5], Danielle L. Mowery[5], Karen L. Olson[16], Lav P. Patel[14], Malarkodi J. Samayamuthu[5], Fernando J. Sanz Vidorreta[17], Emily R. Schriver[18], Petra Schubert[19], Guillaume Verdy[1], Shyam Visweswaran[5], Xuan Wang[5], Griffin M. Weber[5], Zongqi Xia[20], William Yuan[5], Harrison G. Zhang[5], Daniela Zöller[21], Isaac S. Kohane[5], The Consortium for Clinical Characterization of COVID-19 by EHR (4CE)[5¶], Alexandre Boyer[2], Vianney Jouhet[8]

1 IAM Unit, Bordeaux University Hospital, Bordeaux, France, 2 Medical Intensive Care Unit, Bordeaux University Hospital, Bordeaux, France, 3 Inserm Bordeaux Population Health Research Center UMR 1219, Inria BSO, Team SISTM, University of Bordeaux, Bordeaux, France, 4 Department of Biomedical Informatics, Hôpital Necker-Enfants Malade, Assistance Publique Hôpitaux de Paris (APHP), University of Paris, Paris, France, 5 Department of Biomedical Informatics, Harvard Medical School, Boston, Massachusetts, United States of America, 6 Population Health and Data Science, MAVERIC, VA Boston Healthcare System, Boston, Massachusetts, United States of America, 7 INSERM Bordeaux Population Health ERIAS TEAM, Bordeaux University Hospital / ERIAS - Inserm U1219 BPH, Bordeaux, France, 8 Institute of Digitalization in Medicine, Faculty of Medicine and Medical Center, University of Freiburg, Freiburg, Germany, 9 IAM Unit, INSERM Bordeaux Population Health ERIAS TEAM, Bordeaux University Hospital / ERIAS - Inserm U1219 BPH, Bordeaux, France, 10 Department of Learning Health Sciences, University of Michigan, Ann Arbor, Michigan, United States of America, 11 Massachusetts Veterans Epidemiology Research and Information Center (MAVERIC), VA Boston Healthcare System, Boston, Massachusetts, United States of America, 12 Department of Preventive Medicine, Northwestern University, Chicago, Illinois, United States of America, 13 Department of Medicine, Massachusetts General Hospital, Boston, Massachusetts, United States of America, 14 Institute of Medical Biometry and Statistics, Faculty of Medicine and Medical Center, University of Freiburg, Freiburg, Germany, 15 Department of Biostatistics, Epidemiology, and Informatics, University of Pennsylvania Perelman School of Medicine, Philadelphia, Pennsylvania, United States of America, 16 Department of Anaesthesia, National University Health System, Singapore, Singapore, 17 Computational Health Informatics Program, Boston Children's Hospital, Department of Pediatrics, Harvard Medical School, Boston, Massachusetts, United States of America, 18 Department of Internal Medicine, Division of Medical Informatics, University of Kansas Medical Center, Kansas City, Kansas, United States of America, 19 Department of Medicine, David Geffen School of Medicine at UCLA, Los Angeles, California, United States of America, 20 Data Analytics Center, University of Pennsylvania Health System, Philadelphia, Pennsylvania, United States of America, 21 Department of Neurology, University of Pittsburgh, Pittsburgh, Pennsylvania, United States of America

¶ Membership of the author group can be found in the Acknowledgments.
* bertrandmoal@gmail.com



**Data Availability Statement:** All relevant data are available at: https://github.com/covidclinical/ARDS_aggregated_data_Public.

**Funding:** KC is supported by VA MVP000 and CIPHER. DAH is supported by National Institutes of Health (NIH) National Center for Advancing Translational Sciences (NCATS) UL1TR002240. YL is supported by NIH/NCATS U01TR003528 and NIH National Library of Medicine (NLM) 1R01LM013337. MM is supported by Clinical and Translational Science Award (CTSA) UL1 TR001857. DLM is supported by NIH/NCATS CTSA UL1-TR001878 (University of Pennsylvania). LPP is supported by CTSA Award UL1TR002366. SV is supported by NIH/NLM R01LM012095 and NIH/NCATS UL1TR001857. GMW is supported by NIH/NCATS UL1TR002541, NIH/NCATS UL1TR000005, NIH/NLM R01LM013345, and NIH National Human Genome Research Institute (NHGRI) 3U01HG008685-05S2. ZX is supported by NIH National Institute of Neurological Disorders and Stroke (NINDS) R01NS098023. Funders had no role in study design, data collection and analysis, decision to publish, or preparation of the manuscript.

**Competing interests:** The authors have declared that no competing interests exist.

**Abbreviations:** 4CE, Consortium for Clinical Characterization of COVID-19 by EHR; ARDS, acute respiratory distress syndrome; EHR, electronic health records; HS, healthcare systems; ICD, international classification diseases; ICU, intensive care unit; SARS-CoV-2, severe acute respiratory syndrome coronavirus 2.

# Abstract

## Purpose

In young adults (18 to 49 years old), investigation of the acute respiratory distress syndrome (ARDS) after severe acute respiratory syndrome coronavirus 2 (SARS-CoV-2) infection has been limited. We evaluated the risk factors and outcomes of ARDS following infection with SARS-CoV-2 in a young adult population.

## Methods

A retrospective cohort study was conducted between January 1st, 2020 and February 28th, 2021 using patient-level electronic health records (EHR), across 241 United States hospitals and 43 European hospitals participating in the Consortium for Clinical Characterization of COVID-19 by EHR (4CE). To identify the risk factors associated with ARDS, we compared young patients with and without ARDS through a federated analysis. We further compared the outcomes between young and old patients with ARDS.

## Results

Among the 75,377 hospitalized patients with positive SARS-CoV-2 PCR, 1001 young adults presented with ARDS (7.8% of young hospitalized adults). Their mortality rate at 90 days was 16.2% and they presented with a similar complication rate for infection than older adults with ARDS. Peptic ulcer disease, paralysis, obesity, congestive heart failure, valvular disease, diabetes, chronic pulmonary disease and liver disease were associated with a higher risk of ARDS. We described a high prevalence of obesity (53%), hypertension (38%-although not significantly associated with ARDS), and diabetes (32%).

## Conclusion

Trough an innovative method, a large international cohort study of young adults developing ARDS after SARS-CoV-2 infection has been gather. It demonstrated the poor outcomes of this population and associated risk factor.

# Introduction

Acute respiratory distress syndrome (ARDS) [1], is a frequent complication after severe acute respiratory syndrome coronavirus 2 (SARS-CoV-2) infection. According to studies, it appears in 3.4% of the population with a laboratory positive PCR confirmation of infection to the SARS-CoV-2 [2], up to 31% of hospitalized patients [3–5], and 92% of patients admitted to the intensive care unit [4] (ICU).

ARDS has a severe impact on patient outcomes. In a cohort study carried out in New York City on COVID-19 patients, the mortality of ARDS patients reached 39% [4]. ARDS has been frequently associated with long-term disabilities [6–10] and represents a heavy care burden for health systems [11] due to long ICU stays and extended rehabilitation [7, 9].

Age is an important risk factor for developing ARDS [3]. However, young adults (18–49 years old) represented a third of hospitalized patients [12] and a quarter of patients admitted

to the ICU [4]. Based on the Premier Healthcare Database, which includes 1,030 hospitals in the United States, Cunningham et al. [13] reported that 21% of young adults (aged 18 to 34 years) hospitalized with COVID-19 disease were admitted to the ICU and 10% required mechanical ventilation. Similarly, in a separate cohort, young adults represented more than 20% of the patients admitted to ICUs for COVID-19 infection with ARDS [3].

Few studies [12–15] have investigated the young adult population, mostly were single-center analyses, all exclusively in the U.S. population and none focused on ARDS patients. To our knowledge, there have been no specific studies on ARDS after SARS-CoV-2 infection in the young adult population among an international cohort. This may be due to the difficulty in obtaining a large sample of this population. Key questions remain related to the risk factors of ARDS in young adults, and the difference, in terms of outcomes, compared to an older population.

In this study, we investigate the risk of ARDS among young adults hospitalized with COVID-19 using an international cohort from the international Consortium for Clinical Characterization of COVID-19 (4CE) [16–21]. This international consortium collects data from 342 hospitals in 6 countries and develops an innovative federated approach for electronic health records (EHR) analysis.

Through a federated analysis, the objectives were to evaluate the risk factors for developing ARDS following infection with SARS-CoV-2 and hospitalization in young adults and to compare characteristics, care, and outcomes between this population and an older population (greater than 49 years old) who similarly developed ARDS during their COVID-19 hospitalization.

## Patients and methods

The 4CE consortium [16–21] has developed a framework to extract and standardize data directly from the EHRs of participating healthcare systems (HS) and to streamline federated analyses without sharing patient-level data. A common data model for structuring patient-level data was adopted to enable identical analyses across all participating HS. Fig 1 presents the workflow from 4CE data collection to ARDS analysis.

### Common 4CE data collection by HS

As previously described [16], each participating HS were responsible for and obtained ethics approval, as needed, from the appropriate ethics committee at their institution. IRB protocols were reviewed and approved at APHP (IRB00011591, Project CSE-20-29_ClinicalCOVID),

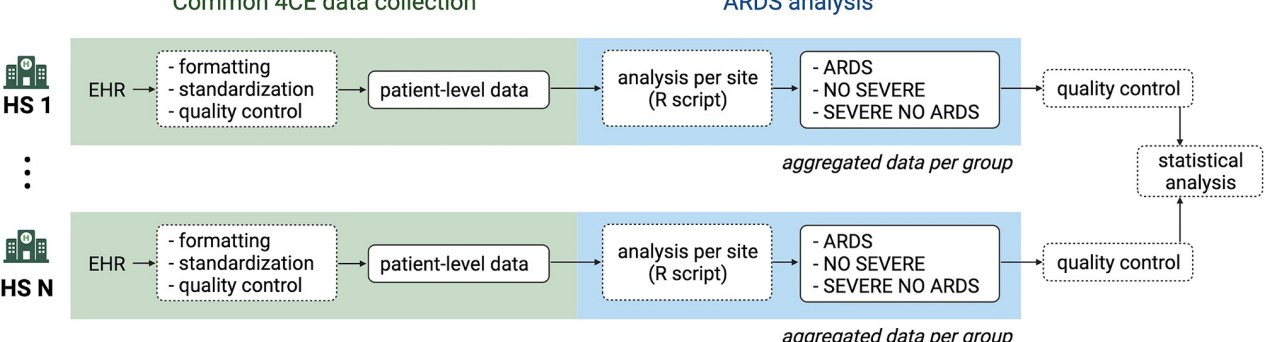

**Fig 1. Study workflow.** From EHR extraction to ARDS analysis on aggregated data (HS: healthcare system).

Bordeaux University Hospital (Registration #CHUBX2020RE0253), Mass General Brigham (IRB#2020P001483), Northwestern University (IRB# STU00212845), University of Kansas (STUDY00146505), University of Freiburg (Application #255/20, Process #210587), and at VA North Atlantic, Southwest, Midwest, Continental, and Pacific (IRB # 3310-x).

The research was determined to be exempt at University of Michigan (IRB# HUM00184357), Beth Israel Deaconess Medical Center (IRB# 2020P000565), University of Pittsburgh (STUDY20070095), and University of Pennsylvania (IRB#842813). University of California Los Angeles determined that this study does not need IRB approval because research using limited data sets does not constitute human subjects research.

**Cohort identification.**   Across each participating HS, we included all hospitalized patients within 7 days before and up to 14 days after a positive PCR SARS-CoV-2 test. The first hospital admission date within this time window was considered day 0 (the index date). Note that although all patients had a positive PCR test near their admission date, it is possible that for some patients the hospitalization was for reasons other than COVID-19.

**Patient-level data collection by HS.**   Patient-level data were collected by HSs, which can represent one or several hospitals. At each HS, data were extracted directly from the EHR and consisted of time to admission and discharge, survival status, sex and age group [18–25, 26–49, 50–69, 70–79, and 80+ years old]. Diagnoses were collected from the first 3 digits of the billing code using international classification disease (ICD) version 10. This 3-digit rollup was adopted to account for finer-grained differences in coding practices across hospitals. Procedures related to endotracheal tube insertion or invasive mechanical ventilation were collected and were denoted as severe procedures [17]. Medications administered were collected at the class level (as per the ATC standard nomenclature [22], S1 Appendix). Severe medication [17] refers to sedatives/anesthetics or treatment for shock (classes: SIANES, SICARDIAC).

All patient-level data were standardized to a common format, then stored and analyzed locally at each HS. Several quality controls were conducted iteratively at each HS to ensure the quality of the data.

## ARDS analysis

**Data aggregation by HS for ARDS analysis.**   Final data extraction was completed on 30<sup>th</sup> August 2021 and included patient hospitalizations occurring from 1<sup>st</sup> January 2020 to 28<sup>th</sup> February 2021. All patients of 18 years or older were included in the analysis. ARDS patients were identified using the ICD10 code, J80—Acute respiratory distress syndrome.

Using patient-level data, each HS ran an R script locally to classify patients into 3 groups as follows:

- ARDS: Patients with an ARDS ICD code

- NO_SEVERE: Patients without an ARDS ICD code, severe medication or severe procedure

- SEVERE_NO_ARDS: Patients with severe medication or severe procedure but without an ARDS ICD code

For the analysis, the cohort was divided into two age groups: patients aged 18 to 49 years and patients older than 49 years (Fig 2). For each group, the number of patients was aggregated in terms of:

- Age, sex, mortality at 90 days after the admission

- Each ICD code, Elixhauser index (23) and complication class (S2 and S3 Appendices)

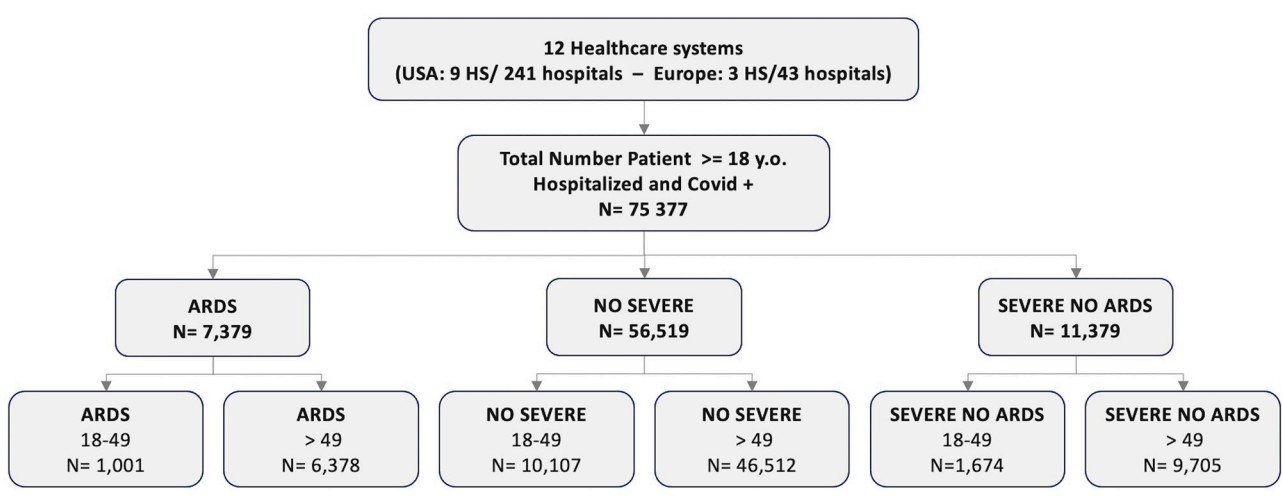

**Fig 2. Flow chart.** Distribution of patients per group (y.o. = years old).

Aggregate data were centrally collected, and several quality controls were executed before pooling the aggregated data together. Descriptive analysis was presented S1 Table.

**Statistical analysis.** Risk factor: comparison between young patients with and without ARDS.

To identify the risk factors associated with an ARDS after SARS-CoV-2 infection and hospitalization, we compared the young patients with ARDS and the young non severe patients. Patients classified in the "SEVERE_NO_ARDS" group were excluded from this analysis.

For comorbidities classified by the Elixhauser Comorbidity Index [23], risk ratios with confidence intervals were calculated from a univariable analysis considering diagnoses recorded between 365 days before (-365) the admission and 90 days after (+90) the admission. First univariable analysis was performed at each HS and aggregated through a random effect meta-analyses to account for heterogeneity between HS. In addition, comorbidities associated with ARDS in this meta univariable analysis and sex were selected for a multivariable analysis. Multivariable analysis was performed at each HS and then aggregated through another meta-analysis with random effect.

Complications and mortality: comparison between young and old adults with ARDS.

The proportion of patients per sex were evaluated and compared between young adults and older adults with ARDS. Complications were identified as novel diagnoses established between the day of admission and +90 days after the admission. To compare complications between young and older patients with ARDS, we performed a univariable analysis and reported estimated risk ratios with confidence intervals. Moreover, mortality was evaluated for both groups at 90 days after the index admission.

Statistical analyses were performed locally at each HS and then aggregated via meta-analysis with the R package metafor [24].

## Results

12 HS participated in the analysis: 9 U.S. HS representing 241 hospitals, two French HS representing 42 hospitals, and one German HS representing 1 hospital (Table 1). 75,377 hospitalized patients with biological confirmation of COVID were included in the analysis.

**Table 1. Name, city, country, number of hospitals per HS, number of beds and inpatient discharges/year per HS.**

| Healthcare System | City | Country | Hospitals | Beds | Inpatient discharges/year |
|---|---|---|---|---|---|
| Assistance Publique—Hôpitaux de Paris | Paris | France | 39 | 20,098 | 1,375,538 |
| Bordeaux University Hospital | Bordeaux | France | 3 | 2,676 | 130,033 |
| Medical Center, University of Freiburg | Freiburg | Germany | 1 | 1,660 | 71,500 |
| Beth Israel Deaconess Medical Center | Boston, MA | USA | 1 | 673 | 40,752 |
| Mass General Brigham (Partners Healthcare) | Boston, MA | USA | 10 | 3,418 | 163,521 |
| University of Pennsylvania | Philadelphia, PA | USA | 5 | 2,469 | 118,188 |
| University of Michigan | Ann Arbor, MI | USA | 3 | 1,000 | 49,008 |
| Northwestern University | Chicago, IL | USA | 10 | 2,234 | 103,279 |
| University of California, LA | Los Angeles, CA | USA | 2 | 786 | 40,526 |
| University of Pittsburgh / UPMC | Pittsburgh, PA | USA | 39 | 8,085 | 369,300 |
| University of Kansas Medical Center | Kansas City, KS | USA | 1 | 794 | 54,659 |
| Veteran affairs | Multiple cities | USA | 170 | 13,801 | 680,687 |

About 7.8% (1001/12,782, HS range: 1.6 to 15%) of hospitalized young adults with COVID developed ARDS compared to 10.2% (6378/62595, HS range: 1.8 to 21.2%) of older patients. Young patients represented 13.4% (1001/7379) of ARDS patients (HS range: 6.5% to 24.5%).

## Risk factors: Comparison between young adults with ARDS and young non severe patients (Table 2)

For the risk factor analysis, young ARDS patients (n = 1001) were compared to young non severe patients (n = 10,107). Among young ARDS patients, 43/1001 (4.3%) were aged between 18 to 25 years old. In an univariable analysis, patients aged 26 to 49 years old had an increased risk of developing ARDS compared to those aged 18 to 25 years old (RR = 2.94; 95% CI: [2.11; 4.1]). Due to the low proportion of patients between 18 to 25 years, age class was not included in the multivariable analysis.

In the multivariable analysis, compared to women, men had a higher risk for developing ARDS (RR = 1.71; 95% CI: [1.20; 2.43]) and the following comorbidities were significantly associated with ARDS: Peptic ulcer disease (RR = 3.66; 95% CI: [2.01; 6.49]), Paralysis (RR = 3.73; 95% CI:[2.52; 5.51]), Obesity (RR = 2.82; 95% CI: [2.06; 3.95]), Congestive heart failure (RR = 2.2; 95% CI: [1.36; 3.57]), Valvular disease (RR = 1.89; 95% CI: [1.08; 3.29]), Diabetes (RR = 1.85; 95% CI: [1.44; 2.38]), Chronic pulmonary disease (RR = 1.62; 95% CI: [1.34; 1.96]) and Liver disease (RR = 1.61; 95% CI: [1.12; 2.31]). Hypertension was not significantly associated (RR = 1.36 [0.98; 1.89]).

Peripheral vascular disease, and renal failure were associated with developing ARDS in univariable analysis, but not in multivariable analysis. AIDS/HIV, alcohol abuse, cancer, drug abuse, hypothyroidism, and psychosis were not associated with higher risk. Nicotine dependency was not associated with a higher risk (p = 0.138).

In the young ARDS population, we observed a high prevalence of comorbidities including obesity 533/1001 (53.3%), diabetes 382/1001 (38.2%), and hypertension 322/1001 (32.2%).

## Complications and mortality: Comparison between young and old adult population with ARDS

6378 patients aged > 49 with ARDS were compared to the young adult population with ARDS. The percentage of males was 67.1% (672/1001) and 75.2% (4797/6378) for the

**Table 2. Number and percentage of patients per age groups, per sex, per Elixhauser comorbidities for young adult patients with ARDS and non severe young adult patients.** Risk ratio associated in uni- and multivariable analysis.

| Variables | ARDS | NO SEVERE | Univariable analysis | | Multivariable analysis | |
|---|---|---|---|---|---|---|
| | ages 18–49 | ages 18–49 | | | | |
| | n = 1001 | n = 10107 | Risk Ratio with CI (95%) | p-value | Risk Ratio with CI (95%) | p-value |
| | n (%) | n (%) | | | | |
| **Age groups, reference: 18 to 25 years old** | | | | | | |
| 18to25 | 43 (4.3) | 1207 (11.9) | 2.9 [2.1; 4.1] | <0.001 | not include | |
| 26to49 | 966 (96.5) | 8900 (88.1) | | | | |
| **Sex, reference: female** | | | | | | |
| female | 327 (32.7) | 4427 (43.8) | 1.7 [1.3; 2.2] | <0.001 | 1.71 [1.2; 2.4] | 0.003 |
| male | 672 (67.1) | 5680 (56.2) | | | | |
| **Comorbidities (Elix Hauser class), ICD code from -365 days before to + 90 days after admission** | | | | | | |
| AIDS/HIV | 12 (1.2) | 121 (1.2) | 1 [0.5; 1.9] | 0.987 | not include | |
| Alcohol abuse | 59 (5.9) | 895 (8.9) | 1 [0.7; 1.4] | 0.92 | not include | |
| Cancer | 37 (3.7) | 280 (2.8) | 1.3 [0.9; 1.7] | 0.164 | not include | |
| Chronic pulmonary disease | 219 (21.9) | 1406 (13.9) | 1.8 [1.6; 2.1] | <0.001 | 1.6 [1.3; 2.0] | <0.001 |
| Congestive heart failure | 143 (14.3) | 532 (5.3) | 3.4 [2.6; 4.4] | <0.001 | 2.2 [1.4; 3.6] | 0.001 |
| Diabetes | 322 (32.2) | 1691 (16.7) | 2.5 [2; 3.1] | <0.001 | 1.9 [1.4; 2.4] | <0.001 |
| Drug abuse | 65 (6.5) | 828 (8.2) | 1 [0.8; 1.3] | 0.997 | not include | |
| Hypertension | 382 (38.2) | 2274 (22.5) | 2.5 [2; 3.2] | <0.001 | 1.4 [0.98; 1.9] | 0.062 |
| Hypothyroidism | 45 (4.5) | 431 (4.3) | 1.4 [1; 2.1] | 0.077 | not include | |
| Liver disease | 179 (17.9) | 960 (9.5) | 2.1 [1.6; 2.8] | <0.001 | 1.6 [1.1; 2.3] | 0.01 |
| Obesity | 533 (53.2) | 2759 (27.3) | 2.9 [2.2; 3.9] | <0.001 | 2.8 [2.0; 4.0] | <0.001 |
| Paralysis | 64 (6.4) | 162 (1.6) | 2.9 [2.3; 3.6] | <0.001 | 3.7 [2.5; 5.5] | <0.001 |
| Peptic ulcer disease | 36 (3.6) | 60 (0.6) | 4.2 [2.9; 6] | <0.001 | 3.7 [2.1; 6.5] | <0.001 |
| Peripheral vascular disease | 37 (3.7) | 184 (1.8) | 2.7 [1.7; 4.2] | <0.001 | 1.2 [0.7; 2.1] | 0.485 |
| Psychoses | 52 (5.2) | 599 (5.9) | 1.1 [0.8; 1.4] | 0.513 | not include | |
| Renal failure | 131 (13.1) | 590 (5.8) | 2.4 [1.9; 2.9] | <0.001 | 1.3 [0.9; 1.8] | 0.158 |
| Valvular disease | 92 (9.2) | 346 (3.4) | 2.9 [2.1; 4] | <0.001 | 1.9 [1.1; 3.3] | 0.025 |

young population and the old population, respectively, without significant difference (p = 0.457).

**Complications (Table 3).** Young ARDS patients had a lower risk of developing the following complications: Acute kidney failure (RR = 0.76; 95% CI: [0.68; 0.85]); cardiac rhythm/conduction disorder (RR = 0.59; 95% CI: [0.47; 0.73]), disorders of fluid, electrolyte and acid-base balance (RR = 0.95; 95% CI: [0.88; 0.99]); and stroke (RR = 0.35; 95% CI: [0.23; 0.53]). However, they had a higher risk of developing pneumonia due to Streptococcus pneumoniae (RR = 1.78; 95% CI:1[1.16; 2.75]), and Streptococcal sepsis (RR = 1.58; 95% CI: [1.08; 2.31]). More than half of the young ARDS patients had Respiratory bacterial superinfection (538/1001 (53.8%)) during their hospitalization. No significant differences were found for the occurrence of pulmonary embolism (p = 0.671), affecting one in 10 patients in both groups with ARDS.

## Mortality

90 days after admission, 16.2% (162/1001) of the young ARDS patients were deceased (HS range [11.2%; 36.8%]). In the older adult population with ARDS patients, the mortality was 41.1% (2619/6378, HS range [24.3%; 76.7%]).

**Table 3. Proportion and associated risk ratio of complication classes for the young compared to old adult with ARDS.**

| Complications | ARDS (ages 18–49) n = 1001 n (%) | ARDS (ages > 49) n = 6378 n (%) | Risk Ratio with CI (95%) | p-value |
|---|---|---|---|---|
| Acute kidney failure | 403 (40.3) | 3431 (53.8) | 0.8 [0.7; 0.9] | <0.001 |
| Cardiac arrest | 57 (5.7) | 455 (7.1) | 1.1 [0.8; 1.5] | 0.691 |
| Cardiac complication | 255 (25.5) | 2195 (34.4) | 0.8 [0.6; 0.9] | 0.01 |
| Cardiac Rhythm/conduction disorder | 310 (31) | 2847 (44.6) | 0.6 [0.5; 0.7] | <0.001 |
| Digestive complication | 393 (39.3) | 2643 (41.4) | 1 [0.9; 1.1] | 0.907 |
| Disorders of fluid, electrolyte and acid-base balance | 546 (54.5) | 3730 (58.5) | 0.9 [0.9; 1] | 0.02 |
| Haematological disorder | 388 (38.8) | 2440 (38.3) | 1 [0.9; 1] | 0.528 |
| Hemodynamic disorder | 271 (27.1) | 1852 (29) | 1 [0.8; 1.1] | 0.573 |
| Arterial embolism and thrombosis | 14 (1.4) | 100 (1.6) | 1.1 [0.6; 1.9] | 0.737 |
| Stroke | 25 (2.5) | 509 (8) | 0.4 [0.2; 0.5] | <0.001 |
| Phlebitis and thrombophlebitis | 180 (18) | 777 (12.2) | 1.3 [1; 1.6] | 0.078 |
| Pulmonary embolism | 105 (10.5) | 695 (10.9) | 1 [0.8; 1.2] | 0.637 |
| Respiratory complication (excluding ARDS) | 857 (85.6) | 5502 (86.3) | 1 [0.9; 1] | 0.202 |
| Pressure ulcer | 115 (11.5) | 818 (12.8) | 1 [0.8; 1.2] | 0.875 |
| Viral reactivation | 29 (2.9) | 177 (2.8) | 1.2 [0.8; 1.8] | 0.356 |
| Infections | | | | |
| Aspergillosis | 26 (2.6) | 164 (2.6) | 0.7 [0.5; 1.2] | 0.179 |
| Candidiasis | 64 (6.4) | 421 (6.6) | 1.2 [0.9; 1.5] | 0.182 |
| Other fungal infection | 21 (2.1) | 111 (1.7) | 1.1 [0.6; 1.9] | 0.768 |
| Bacterial infection | 528 (52.7) | 3366 (52.8) | 0.9 [0.9; 1] | 0.187 |
| Bacterial intestinal infection | 41 (4.1) | 242 (3.8) | 1.2 [0.9; 1.6] | 0.299 |
| Respiratory bacterial superinfection | 538 (53.7) | 3507 (55) | 1 [0.9; 1.1] | 0.869 |
| Pneumonia due to Streptococcus pneumoniae | 34 (3.4) | 107 (1.7) | 1.8 [1.2; 2.7] | 0.009 |
| Streptococcal sepsis | 41 (4.1) | 145 (2.3) | 1.6 [1.1; 2.3] | 0.018 |

### Data

The aggregated data per site are available here. Sites were anonymized.

### Discussion

In a large international EHR-based cohort, we employed a novel federated approach including 241 hospitals in the United States and 43 in Europe, to describe comorbidities, complications, and mortality of young adults developing ARDS after SARS-CoV-2 infection. Even though young patients with ARDS represent a small proportion of hospitalized patients with COVID (HS range: [0.4%; 3.3%]), we were able to gather a large cohort thanks to this innovative method and demonstrated the poor outcome of young ARDS patients with notable mortality (16.2%).

### Mortality and complications

Independently on the etiology, in-hospital mortality for ARDS patients has been reported to be between 30 to 40% [7, 25, 26]. Mortality at 30 days for ARDS patients of any age with COVID-19 was reported at 39% [4] and corresponds to the mortality for the older ARDS population in our study. The young ARDS population's mortality at 90 days was smaller, around 16.2% with large variability between HS [11.2; 36.8%]. Importantly, it was not possible to assess

the attributable COVID-19 mortality from our data. However the mortality appeared high for this young population; in a 2018 study conducted in France, all-cause mortality of ICU patients in the same age range was estimated to be less than 10% [27]. The relatively higher risk of developing pneumonia due to Streptococcus pneumoniae and Streptococcal sepsis in young adults is probably related to their greater survival rate compared to older patients. The high frequency of complications in this young population emphasizes the major impact of ARDS on poor outcomes and mortality.

## Risk factors

Although the proportion of the general population is low, ARDS appears in 7.8% of young hospitalized adults with COVID. These percentages are in agreement with those reported by Cummings et al. [3] and Cunningham et al. [13]. Among those young ARDS patients only 4,3% were aged between 18- and 25-years old. Patients developing ARDS in this young adult population had a high prevalence of obesity (53%), hypertension (38%) and diabetes (32%).

A limitation of relying on billing codes to identify comorbidities is the challenge of accurately distinguishing comorbidities from complications. In our analysis, comorbidities were considered as those diagnoses from billing codes assigned up to one year before and up to 90 days after the admission. In electronic health records, each code is attached to one specific hospitalization visit. For patients with prior hospitalizations, comorbidities are easily identified with the codes attached to those previous hospitalization. However, for patient without previous hospitalization, the fact that electronic health records do not contains any code, do not mean that patient did not have comorbidities. For example, an obese patient without prior hospitalization would be identified as "obese"only if we take into account the code associated with the index hospitalization. This approach is more sensitive, but it can lead to considering complications as comorbidities. It is particularly true for peptic ulcer disease or paralysis which was identified as a comorbidity associated with ARDS but which is also known to be a common complication of mechanical ventilation [28, 29] or prolonged ICU admission. We perform a complementary univariable analysis on the sub population who had previous hospital visits and considering only the ICD code related to those previous visits as comorbidities (one year and– 14 days before the admission). In this univariable analysis presented in S2 Table, ARDS was associated with the presence of peptic ulcer disease or paralysis in a previous hospitalization, which explained our choice to keep both in the main multivariable analysis that means considering them as comorbidities. "Paralysis" regroups is related a large diversity of diagnoses. including encephalitis, myelitis and encephalomyelitis, hereditary ataxia, cerebral palsy, hemiplegia and hemiparesis, paraplegia (paraparesis) and quadriplegia (quadriparesis), and other paralytic syndromes (S2 Appendix); but a common co-occurrence is reduced lung capacity which could contribute to its association with ARDS. The association with peptic ulcer as comorbidities remains unclear and requires additional investigations.

Obesity has been identified as a risk factor for poor outcome for ARDS [30] and for SARS-CoV-2 infection [3, 14, 15, 31] and it also appears in this analysis as a risk factor in this young adult population. Diabetes has a controversial association with ARDS [32–34] but appears in this population as a risk factor and has also been associated with the severity of SARS-CoV-2 infection in other studies [3, 14, 15, 31]. Despite its association with poor outcomes in several cohorts of COVID-19 patients [2, 15], hypertension was not significantly associated with ARDS in our study, possibly due to the choice of the variable included in the multivariable analysis and/or a lack of power.

Congestive heart failure, valvular disease, chronic liver disease, and chronic pulmonary disease are not associated with ARDS in the literature, however, their associations with

COVID-19 have been identified as a risk factor for poor outcomes [3, 14, 15, 31]. Through our analysis, it seems that most of the comorbidities associated with ARDS in the young adult population are similar to the ones associated with poor outcomes after SARS-CoV-2 infection in the general population. However, for most of them, it is unclear whether they are truly related to the onset of ARDS or just general comorbidities. Further analysis needs to be carried out to eliminate confounding factors and better understand the potential mechanisms of those associations.

## Limitations

Our major limitation is that group membership, comorbidities, and complication analyses are based on billing codes, procedures, and medications directly extracted from EHR. Variation in billing coding practices, especially across international healthcare systems, may result in missing data and related biases [35]. However, multiple quality controls have been established to reduce those potential biases. For the detection of ARDS patients, a correct sensibility is expected as billing code is related to reimbursement in most countries and ARDS is associated with heavy care. Regarding the relation between ARDS and COVID-19 infection, patients included in our analysis had positive reverse transcription PCR tests for SARS-CoV-2 infection 7 days before to 14 days after the date of admission. This inclusion criterion allows us to ensure that included patients had the COVID-19 infection at least at the beginning of the hospitalization. Even if it is not possible to establish a clear temporal relationship or causality between ARDS and COVID-19 with ICD codes, it would be extremely rare that the development of ARDS during the hospitalization of a COVID-19 positive patient had no relation to COVID-19 infection. It is possible that COVID-19 infection was not the primary cause of the ARDS but most likely had an impact on the ARDS development.

To identify comorbidities associated with ARDS following hospitalization with COVID, a comparison was performed considering only non severe patients. Patients with mechanical ventilation, sedatives/anesthetics, or treatment for shock but without ARDS code were not included, which could generate a selection bias. This choice was conducted to eliminate potential miscoded ARDS patients and patients with severe disease or care not related to SARS-CoV-2 infection but with a concomitant infection. Those patients could have been included in the ARDS population, but the objective of this study was to focus precisely on ARDS patients, and this grouping would have resulted in a significant measurement bias. Especially because the number of young SEVERE_NO_ARDS patients is greater than the one of ARDS patients. In addition, we believe that the descriptive analysis of the SEVERE_NO_ARDS brings credit to this choice (S1 Table). Compared to the other groups, SEVERE_NO_ARDS population had the higher percentage of women (52.2%) and of patients with previous contact with the healthcare system (72%). In addition, 15.1% of those patients had a billing code associated with pregnancy and 36.1% with long-term drug therapy. These results suggest that the COVID-19 infection was simply concomitant but not the main cause of these hospitalizations.

Treatment like the use of mechanical ventilation, ECMO or even ICU admission were not collected. The collection of treatment data, described by specific codes in EHR, has proven to be too partial and largely heterogeneous between health systems (even from the same country) to be collected. It also appears that the accuracy of ICU admission in EHR data was poor. It was particularly true at the beginning of the pandemic, where hallways were converted into ad hoc ICUs to support the surge of sick patients, without notification in the chart. This issue has already been discussed in a previous article from the consortium [17].

More detailed analysis on age's threshold was not possible because age was intentionally not collected by the 4CE consortium as a continuous variable. This choice was made to ensure

greater security/de-identification on the data collection process which allowed for an easier regulatory process for international aggregated data sharing.

## Conclusion

We federated a large EHR-based international cohort of young adults developing ARDS after COVID-19. ARDS appears in 7.8% of hospitalized young patients with COVID and was associated with high mortality (16.2%). Young adults developing ARDS presented a high prevalence of comorbidities, particularly obesity, hypertension (although not being associated with ARDS), and diabetes. ARDS development was associated with peptic ulcer disease, paralysis, obesity, congestive heart failure, valvular disease, diabetes, chronic pulmonary disease, and liver disease.

## Supporting information

**S1 Appendix. Medication class.**
(DOCX)

**S2 Appendix. Elixhauser comorbidities.**
(DOCX)

**S3 Appendix. Complication classification.**
(DOCX)

**S1 Table. Number and percentage of patients per age groups, per sex, per Elixhauser comorbidities for all groups.**
(DOCX)

**S2 Table. Number and percentage of patients per Elixhauser comorbidities for young adult patients with ARDS and non severe young adult patients.** Risk ratio associated in univariable analysis for the sub population which had previous hospital visits and considering only the ICD code related to those previous visits (one year and– 14 before the admission).
(DOCX)

## Acknowledgments

The Consortium for Clinical Characterization of COVID-19 by EHR (4CE)

Lead author: Isaac S Kohane MD, PhD

James R Aaron MHA[1], Giuseppe Agapito PhD[2], Adem Albayrak[3], Giuseppe Albi MS[4], Mario Alessiani MD, FACS[5], Anna Alloni PhD[6], Danilo F Amendola MSc[7], François Angoulvant MD,PhD[8], Li L.L.J Anthony[9], Bruce J Aronow PhD[10], Fatima Ashraf MS[11], Andrew Atz MD[12], Paul Avillach MD, PhD[13], Vidul Ayakulangara Panickan MS[13], Paula S Azevedo MD, PhD[14], James Balshi[15], Ashley Batugo BS[16], Brett K Beaulieu-Jones PhD[13], Brendin R Beaulieu-Jones MD, MBA[13], Douglas S Bell[17], Antonio Bellasi MD, PhD[18], Riccardo Bellazzi MS, PhD[4], Vincent Benoit PhD[19], Michele Beraghi MS[20], José Luis Bernal-Sobrino MS[21], Mélodie Bernaux[22], Romain Bey[19], Surbhi Bhatnagar PhD[23], Alvar Blanco-Martínez MS[21], Martin Boeker[24], Clara-Lea Bonzel MSc[13], John Booth MSc[25], Silvano Bosari Prof.[26], Florence T Bourgeois MD, MPH[27], Robert L Bradford[28], Gabriel A Brat MD[13], Stéphane Bréant[29], Nicholas W Brown MEng[13], Raffaele Bruno MD[30], William A Bryant PhD[25], Mauro Bucalo MS[6], Emily Bucholz MD, PhD, MPH[31], Anita Burgun[32], Tianxi Cai ScD[13], Mario Cannataro M.Sc.[33], Aldo Carmona[34], Anna Maria Cattelan MD[35], Charlotte Caucheteux[36], Julien Champ[37], Krista Y Chen BS[38], Jin Chen PhD[39], Luca Chiovato MD, PhD[40], Lorenzo Chiudinelli PhD[41], Kelly

Cho PhD, MPH[42], James J Cimino MD[43], Tiago K Colicchio PhD, MBA[43], Sylvie Cormont[29], Sébastien Cossin[44], Jean B Craig PhD[45], Juan Luis Cruz-Bermúdez PhD[21], Jaime Cruz-Rojo MD[21], Arianna Dagliati MS, PhD[46], Mohamad Daniar MSIS[47], Christel Daniel[48], Priyam Das PhD[13], Batsal Devkota[49], Audrey Dionne MD[31], Rui Duan PhD[50], Julien Dubiel[29], Scott L DuVall PhD[51], Loic Esteve[52], Hossein Estiri PhD[53], Shirley Fan[54], Robert W Follett BS[17], Thomas Ganslandt MD[55], Noelia García-Barrio MS[21], Lana X Garmire PhD[56], Nils Gehlenborg[13], Emily J Getzen MS[57], Alon Geva MD, MPH[58], Tomás González González MD[21], Tobias Gradinger MD, BSc[55], Alexandre Gramfort[36], Romain Griffier[44], Nicolas Griffon[48], Olivier Grisel[36], Alba Gutiérrez-Sacristán PhD[13], pietro h guzzi PhD[59], Larry Han PhD[50], David A Hanauer MD, MS[60], Christian Haverkamp MD[61], Derek Y Hazard MSc[62], Bing He PhD[56], Darren W Henderson BS[1], Martin Hilka[29], Yuk-Lam Ho MPH[63], John H Holmes MS, PhD[64,16], Jacqueline P Honerlaw RN, MPH[63], Chuan Hong PhD[65,13], Kenneth M Huling HS[13], Meghan R Hutch BS[66], Richard W Issitt DClinP[25], Anne Sophie Jannot[67], Vianney Jouhet MD,PhD[44], Ramakanth Kavuluru PhD[68], Mark S Keller[13], Chris J Kennedy PhD[69], Kate F Kernan MD[70], Daniel A Key BEng[25], Katie Kirchoff MSHI[71], Jeffrey G Klann MEng, PhD[53], Isaac S Kohane MD, PhD[13], Ian D Krantz[72], Detlef Kraska Dr.[73], Ashok K Krishnamurthy PhD[74], Sehi L'Yi PhD[13], Trang T Le PhD[64], Judith Leblanc[75], Guillaume Lemaitre[36], Leslie Lenert MD, MS[45], Damien Leprovost[76], Molei Liu PhD[77], Ne Hooi Will Loh MBBS[78], Qi Long PhD[79], Sara Lozano-Zahonero PhD[80], Yuan Luo PhD[66], Kristine E Lynch PhD[51], Sadiqa Mahmood[3], Sarah E Maidlow AA[81], Adeline Makoudjou MD[62], Simran Makwana MS[13], Alberto Malovini PhD[82], Kenneth D Mandl MD, MPH[83], Chengsheng Mao PhD[66], Anupama Maram MS[84], Monika Maripuri MBBS, MPH[63], Patricia Martel[85], Marcelo R Martins MSc[86], Jayson S Marwaha MD[87], Aaron J Masino PhD[88], Maria Mazzitelli MD, PhD[35], Diego R Mazzotti PhD[89], Arthur Mensch[90], Marianna Milano PhD[91], Marcos F Minicucci MD, PhD[92], Bertrand Moal MD, PhD[93], Taha Mohseni Ahooyi PhD[94], Jason H Moore PhD[95], Cinta Moraleda MD, PhD[96], Jeffrey S Morris[97], Michele Morris BA[98], Karyn L Moshal[99], Sajad Mousavi PhD[13], Danielle L Mowery PhD[64], Douglas A Murad[17], Shawn N Murphy MD, PhD[100], Thomas P Naughton BA[101], Carlos Tadeu Breda Neto[7], Antoine Neuraz MD, PhD[102], Jane Newburger MD, MPH[31], Kee Yuan Ngiam MBBS, FRCS[103], Wanjiku FM Njoroge MD[104], James B Norman[13], Jihad Obeid MD, FAMIA[45], Marina P Okoshi PhD[92], Karen L Olson PhD[105], Gilbert S. Omenn MD, PhD[106], Nina Orlova[29], Brian D Ostasiewski BS[107], Nathan P Palmer PhD[13], Nicolas Paris[29], Lav P Patel MS[108], Miguel Pedrera-Jiménez MS[21], Ashley C Pfaff MD[109], Emily R Pfaff PhD[110], Danielle Pillion MS[13], Sara Pizzimenti MS[26], Tanu Priya BS[111], Hans U Prokosch[112], Robson A Prudente PhD[113], Andrea Prunotto PhD[80], Víctor Quirós-González MS[21], Rachel B Ramoni[114], Maryna Raskin[3], Siegbert Rieg MD[115], Gustavo Roig-Domínguez MS[21], Pablo Rojo MD,PhD[96], Paula Rubio-Mayo MS[21], Paolo Sacchi MD[30], Carlos Sáez PhD[116], Elisa Salamanca[29], Malarkodi Jebathilagam Samayamuthu MD[98], L. Nelson Sanchez-Pinto MD, MBI[117], Arnaud Sandrin[29], Nandhini Santhanam MSc[55], Janaina C.C Santos MS[118], Fernando J Sanz Vidorreta[17], Maria Savino MS[119], Emily R Schriver MS[120], Petra Schubert MPH[63], Juergen Schuettler[121], Luigia Scudeller MD, MSc[26], Neil J Sebire MD, FRCPath[122], Pablo Serrano-Balazote MD,MS[21], Patricia Serre[29], Arnaud Serret-Larmande MD[123], Mohsin Shah MSc[25], Zahra Shakeri Hossein Abad PhD[124], Domenick Silvio[125], Piotr Sliz[126], Jiyeon Son MD[127], Charles Sonday[128], Andrew M South MD, MS[129], Francesca Sperotto MD, PhD[31], Anastasia Spiridou PhD[25], Zachary H. Strasser MD[53], Amelia LM Tan BSc, PhD[13], Bryce W.Q. Tan MBBS[130], Byorn W.L. Tan MBBS[130], Suzana E Tanni PhD[92], Deanne M Taylor PhD[131], Ana I Terriza-Torres MS[21], Valentina Tibollo MS[82], Patric Tippmann MSc[132], Emma MS Toh[133], Carlo Torti PhD[134], Enrico M Trecarichi PhD[134], Andrew K Vallejos[135], Gael Varoquaux[136], Margaret E Vella MPH[13], Guillaume Verdy MSc[93], Jill-Jênn Vie[137], Shyam Visweswaran MD, PhD[98], Michele Vitacca MD, PhD[138], Kavishwar B Wagholikar

MBBS, PhD[139], Lemuel R Waitman[140], Xuan Wang PhD[13], Demian Wassermann[36], Griffin M Weber MD, PhD[13], Martin Wolkewitz PhD[132], Scott Wong[130], Zongqi Xia MD, PhD[141], Xin Xiong MS[50], Ye Ye BMED, MSPH, PhD[142], Nadir Yehya MD, MSCE[143], William Yuan PhD[13], Joany M Zachariasse MD, PhD[13], Janet J Zahner BS[144], Alberto Zambelli[145], Harrison G Zhang BA[13], Daniela Zöller PhD[80], Valentina Zuccaro MD[30], Chiara Zucco PhD[91]

[1]Department of Biomedical Informatics, University of Kentucky, Lexington, KY, United States. [2]Department of Legal, Economic and Social Sciences, University Magna Graecia of Catanzaro, Italy, Catanzaro, Italy. [3]Health Catalyst, INC., Cambridge, MA, United States. [4]Department of Electrical, Computer and Biomedical Engineering, University of Pavia, Italy, Pavia, Italy. [5]Department of Surgery, ASST Pavia, Lombardia Region Health System, Pavia, Italy. [6]BIOMERIS (BIOMedical Research Informatics Solutions), Pavia, Italy. [7]Clinical Research Unit of Botucatu Medical School, São Paulo State University, Botucatu, Brazil, Clinical Research Unit of Botucatu Medical School, São Paulo State University, Botucatu, Brazil, Botucatu, Brazil. [8]Pediatric emergency Department, Hôpital Necker-Enfants Malades, Assistance Public-Hôpitaux de Paris, Paris, Paris, France. [9]National Center for Infectious Diseases, Tan Tock Seng Hospital, Singapore, Singapore, Singapore. [10]Departments of Biomedical Informatics, Pediatrics, Cincinnati Children's Hospital Medical Center, University of Cincinnati, Cincinnati, OH, United States. [11]BIG-ARC, The University of Texas Health Science Center at Houston, School of Biomedical Informatics, Houston, TX, United States. [12]Department of Pediatrics, Medical University of South Carolina, Charleston, SC, United States. [13]Department of Biomedical Informatics, Harvard Medical School, Boston, MA, United States. [14]Internal Medicine Department, Botucatu Medical School, São Paulo State University, Botucatu, Brazil, Botucatu, Brazil. [15]Department of Surgery, St. Luke's University Health Network, Bethlehem, PA, Bethlehem, PA, United States. [16]Institute for Biomedical Informatics, University of Pennsylvania Perelman School of Medicine, Philadelphia, PA, United States. [17]Department of Medicine, David Geffen School of Medicine at UCLA, Los Angeles, CA, United States. [18]Department of Medicine, Division of Nephrology, Ente Ospedaliero Cantonale, Lugano, Switzerland, Lugano, Switzerland. [19]IT Department, Innovation & Data, APHP Greater Paris University Hospital, Paris, France. [20]IT Department, ASST Pavia, Voghera, Italy. [21]Health Informatics, Hospital Universitario 12 de Octubre, Madrid, Spain, Madrid, Spain. [22]Strategy and Transformation Department, APHP Greater Paris University Hospital, Paris, France. [23]Department of Biomedical Informatics, Cincinnati Children's Hospital Medical Center, Cincinnati, OH, United States. [24]Technical University of Munich, Munich, Germany. [25]Digital Research, Informatics and Virtual Environments (DRIVE), Great Ormond Street Hospital for Children, UK, London, United Kingdom. [26]Scientific Direction, IRCCS Ca' Granda Ospedale Maggiore Policlinico di Milano, Milan, Italy. [27]Department of Pediatrics, Harvard Medical School, Boston, MA, United States. [28]North Carolina Translational and Clinical Sciences (NC TraCS) Institute, UNC Chapel Hill, Chapel Hill, NC, United States. [29]IT department, Innovation & Data, APHP Greater Paris University Hospital, Paris, France. [30]Division of Infectious Diseases I, Fondazione I.R.C.C.S. Policlinico San Matteo, Italy, Pavia, Italy. [31]Department of Cardiology, Boston Children's Hospital, Harvard Medical School, Boston, MA, United States. [32]Department of Biomedical Informatics, HEGP, APHP Greater Paris University Hospital, Paris, France. [33]Department of Medical and Surgical Sciences, Data Analytics Research Center, University Magna Graecia of Catanzaro, Italy, Catanzaro, Italy. [34]Department of Anesthesia, St. Luke's University Health Network, Bethlehem, PA, Bethlehem, PA, United States. [35]Dipartimento di Medicina dei Sistemi, Infectious and Tropical Disease Unit, Padua University Hospital, Padua, Italy. [36]Université Paris-Saclay, Inria, CEA, Palaiseau, France. [37]INRIA Sophia-Antipolis–ZENITH team, LIRMM, Montpellier, France, Montpellier, France. [38]Computational Health Informatics Program, Boston Children's Hospital, Boston, MA, United States.

[39]Department of Internal Medicine, University of Kentucky, Lexington, KY, United States. [40]Unit of Internal Medicine and Endocrinology, Istituti Clinici Scientifici Maugeri SpA SB IRCCS, Pavia, Italy. [41]UOC Ricerca, Innovazione e Brand reputation, ASST Papa Giovanni XXIII, Bergamo, Bergamo, Italy. [42]Population Health and Data Science, MAVERIC, VA Boston Healthcare System, Boston, MA, United States. [43]Informatics Institute, University of Alabama at Birmingham, Birmingham, AL, United States. [44]IAM unit, INSERM Bordeaux Population Health ERIAS TEAM, Bordeaux University Hospital / ERIAS—Inserm U1219 BPH, Bordeaux, France. [45]Biomedical Informatics Center, Medical University of South Carolina, Charleston, SC, United States. [46]Department of Electrical Computer and Biomedical Engineering, University of Pavia, Italy, Pavia, Italy. [47]Clinical Research Informatics, Boston Children's Hospital, Boston, MA, United States. [48]IT department, Innovation & Data (APHP), UMRS1142 (INSERM), APHP Greater Paris University Hospital, INSERM, Paris, France. [49]Department of Biomedical and Health Informatics, Children's Hospital of Philadelphia, Philadelphia, PA, United States. [50]Department of Biostatistics, Harvard T.H. Chan School of Public Health, Boston, MA, United States. [51]VA Informatics and Computing Infrastructure, VA Salt Lake City Health Care System, Salt Lake City, UT, United States. [52]SED/SIERRA, Inria Centre de Paris, Paris, France. [53]Department of Medicine, Massachusetts General Hospital, Boston, MA, United States. [54]Health Information Technology & Services, University of Michigan, Ann Arbor, MI, United States. [55]Heinrich-Lanz-Center for Digital Health, University Medicine Mannheim, Heidelberg University, Mannheim, Germany. [56]Department of Computational Biology and Bioinformatics, University of Michigan, Ann Arbor, MI, United States. [57]Biostatistics, Perelman School of Medicine at the University of Pennsylvania, Philadelphia, PA, United States. [58]Department of Anesthesiology, Critical Care, and Pain Medicine and Computational Health Informatics Program, Boston Children's Hospital, Boston, MA, United States. [59]Department of Surgical Medical Sciences, University of Catanzaro, Catanzaro, Italy. [60]Department of Learning Health Sciences, University of Michigan Medical School, Ann Arbor, MI, Ann Arbor, MI, United States. [61]Institute of Digitalization in Medicine, Faculty of Medicine and Medical Center, University of Freiburg, Freiburg, Germany. [62]Institute of Medical Biometry and Statistics, Faculty of Medicine and Medical Center, University of Freiburg, Freiburg, Germany. [63]Massachusetts Veterans Epidemiology Research and Information Center (MAVERIC), VA Boston Healthcare System, Boston, MA, United States. [64]Department of Biostatistics, Epidemiology, and Informatics, University of Pennsylvania Perelman School of Medicine, Philadelphia, PA, United States. [65]Department of Biostatistics and Bioinformatics, Duke University, Durham, NC, United States. [66]Department of Preventive Medicine, Northwestern University, Chicago, IL, United States. [67]Department of Biomedical Informatics, HEGP, APHP Greater Paris University Hospital, Paris, France. [68]Division of Biomedical Informatics (Department of Internal Medicine), University of Kentucky, Lexington, KY, United States. [69]Center for Precision Psychiatry, Massachusetts General Hospital, Boston, MA, United States. [70]Department of Critical Care Medicine, Children's Hospital of PIttsburgh, Pittsburgh, PA, United States. [71]Medical University of South Carolina, Charleston, SC, United States. [72]Department of Pediatrics, Division of Human Genetics, The Children's Hospital of Philadelphia and the Perelman School of Medicine at the University of Pennsylvania, Philadelphia, PA, United States. [73]Center for Medical Information and Communication Technology, University Hospital Erlangen, Erlangen, Germany. [74]Renaissance Computing Institute/Department of Computer Science, University of North Carolina, Chapel Hill, Chapel Hill, NC, United States. [75]Clinical Research Unit, Saint Antoine Hospital, APHP Greater Paris University Hospital, Paris, France. [76]Clevy.io, Paris, France. [77]Department of Biostatistics, Harvard T. H. Chan School of Public Health, Boston, MA, United States. [78]Department of Anaesthesia, National University Health System, Singapore, Singapore, Singapore. [79]Department of Biostatistics,

Epidemiology and Informatics, University of Pennsylvania Perelman School of Medicine, Philadelphia, PA, United States. [80]Institute of Medical Biometry and Statistics, Faculty of Medicine and Medical Center, University of Freiburg, Freiburg, Freiburg, Germany. [81]Michigan Institute for Clinical and Health Research (MICHR) Informatics, University of Michigan, Ann Arbor, MI, United States. [82]Laboratory of Informatics and Systems Engineering for Clinical Research, Istituti Clinici Scientifici Maugeri SpA SB IRCCS, Pavia, Italy., Pavia, Italy. [83]Computational Health Informatics Program, Boston Children's Hospital, Boston, MA, United States. [84]Harvard Catalyst, Harvard Medical School, Boston, MA, United States. [85]Clinical Research Unit, Paris Saclay, APHP Greater Paris University Hospital, Boulogne-Billancourt, France. [86]Medical Informatics Center, Hospital das Clínicas, Faculty of Medicine of Botucatu, Clinics hospital of the Botucatu Medical School, São Paulo State University, Botucatu, Brazil, Botucatu, Brazil. [87]Department of Surgery, Beth Israel Deaconess Medical Center, Boston, MA, United States. [88]Department of Anesthesiology and Critical Care, Children's Hospital of Philadelphia, Philadelphia, PA, United States. [89]Department of Internal Medicine, Division of Medical Informatics, University of Kansas Medical Center, Kansas City, KS, United States. [90]ENS, PSL University, Paris, France. [91]Department of Medical and Surgical Sciences, University Magna Graecia of Catanzaro, Italy, Catanzaro, Italy. [92]Internal Medicine Department of Botucatu Medical School, São Paulo State University, Botucatu, Brazil, Botucatu, Brazil. [93]IAM unit, Bordeaux University Hospital, Bordeaux, France. [94]Department of Biomedical Health Informatics, Children's Hospital of Philadelphia, Philadelphia, PA, United States. [95]Department of Computational Biomedicine, Cedars-Sinai Medical Center, West Hollywood, United States. [96]Pediatric Infectious Disease Department, Hospital Universitario 12 de Octubre, Madrid, Spain, Madrid, Spain. [97]Department of Biostatistics, Epidemiology, and Informatics, Institute for Biomedical Informatics, University of Pennsylvania Perelman School of Medicine, Berwyn, United States. [98]Department of Biomedical Informatics, University of Pittsburgh, Pittsburgh, PA, United States. [99]Department of Infectious Diseases, Great Ormond Street Hospital for Children, UK, London, United Kingdom. [100]Department of Neurology, Massachusetts General Hospital, Boston, MA, United States. [101]Harvard Catalyst | The Harvard Clinical and Translational Science Center, Harvard Medical School, Boston, MA, United States. [102]Department of biomedical informatics, Hôpital Necker-Enfants Malade, Assistance Publique Hôpitaux de Paris (APHP), University of Paris, Paris, France. [103]Department of Biomedical informatics, WiSDM, National University Health System Singapore, Singapore, Singapore. [104]Department of Psychiatry, University of Pennsylvania Perelman School of Medicine, Philadelphia, PA, United States. [105]Computational Health Informatics Program and Department of Pediatrics, Boston Children's Hospital and Harvard Medical School, Boston, MA, United States. [106]Dept of Computational Medicine & Bioinformatics, Internal Medicine, Human Genetics, and School of Public Health, University of Michigan, Ann Arbor, MI, United States. [107]CTSI, WFBMI, Wake Forest School of Medicine, Winston Salem, NC, United States. [108]Department of Internal Medicine, Division of Medical Informatics, University Of Kansas Medical Center, Kansas City, KS, United States. [109]Department of Surgery, Beth Israel Deaconess Medical Center, Harvard Medical School, Boston, MA, United States. [110]NC TraCS Institute, UNC Chapel Hill, Chapel Hill, NC, United States. [111]Department of Preventive Medicine, Northwestern University Feinberg School of Medicine, Chicago, IL, United States. [112]Department of Medical Informatics, University of Erlangen-Nürnberg, Erlangen, Germany. [113]Clinical Research Unit São Paulo State University, Brazil, Clinical Research Unit São Paulo State University, Brazil, Botucatu, Brazil. [114]Office of Research and Development, Department of Veterans Affairs, Department of Veterans Affairs, Washington, DC, United States. [115]Division of Infectious Diseases, Department of Medicine II, Medical Center–University of Freiburg, Faculty of Medicine, Freiburg, Germany. [116]Biomedical Data Science Lab, ITACA

Institute, Universitat Politècnica de València, Spain, Valencia, Spain. [117]Department of Pediatrics (Critical Care), Northwestern University Feinberg School of Medicine, Chicago, IL, United States. [118]Nurse departament of FMB—medicine school of Botucatu, Clinical Research Unit of Botucatu Medical School, São Paulo State University, Botucatu, Brazil, Botucatu, Brazil. [119]ASST Pavia, Lombardia Region Health System, Management Engineer, Direction, Pavia, Italy. [120]Data Analytics Center, University of Pennsylvania Health System, Philadelphia, PA, United States. [121]Department of Anesthesiology, University Hospital Erlangen, FAU Erlangen-Nürnberg, Germany, Erlangen, Germany. [122]Digital Research, Informatics and Virtual Environments (DRIVE), Great Ormond Street Hospital for Children NIHR BRC, UK, London, United Kingdom. [123]Department of Biostatistics and Biomedical Informatics, Hôpital Saint-Louis, APHP Greater Paris University Hospital, Paris University, Paris, France. [124]Dalla Lana School of Public Health, University of Toronto, Toronto, Canada. [125]MICHR Informatics, University of Michigan, Ann Arbor, MI, United States. [126]CHIP, Boston Children's Hospital, Boston, MA, United States. [127]Department of Neurology, University of Pittsburgh Medical Center, Pittsburgh, PA, United States. [128]Critical Care Medicine, Department of Medicine, St. Luke's University Health Network, Bethlehem, PA, Bethlehem, PA, United States. [129]Department of Pediatrics-Section of Nephrology, Brenner Children's, Wake Forest University School of Medicine, Winston Salem, NC, United States. [130]Department of Medicine, National University Hospital, Singapore, Singapore, Singapore. [131]Department of Biomedical Health Informatics and the Department of Pediatrics, The Children's Hospital of Philadelphia and the University of Pennsylvania Perelman Medical School, Philadelphia, PA, United States. [132]Institute of Medical Biometry and Statistics, Institute of Medical Biometry and Statistics, Medical Center, University of Freiburg, Freiburg, Germany. [133]Yong Loo Lin School of Medicine, National University of Singapore, Singapore, Singapore. [134]Department of Medical and Surgical Sciences, Infectious and Tropical Disease Unit, University Magna Graecia of Catanzaro, Italy, Catanzaro, Italy. [135]Clinical & Translational Science Institute, Medical College of Wisconsin, Milwaukee, United States. [136]Université Paris-Saclay, Inria, CEA, Montréal Neurological Institute, McGill University, Palaiseau, France. [137]SequeL, Inria Lille, Villeneuve-d'Ascq, France. [138]Respiratory Department, ICS S. Maugeri IRCCS Pavia Italy, Lumezzane (Bs), ITALY. [139]Department of Medicine, Massachusetts General Hospital, Boston, MA, USA. [140]Department of Health Management and Informatics, University of Missouri, Columbia. MO, Columbia, MO, United States. [141]Department of Neurology, University of Pittsburgh, Pittsburgh, PA, United States. [142]Department of Veterans Affairs, 1100 First Street, NW, Washington, DC 20420, University of Pittsburgh, Pittsburgh, PA, United States. [143]Department of Anesthesiology and Critical Care Medicine, Children's Hospital of Philadelphia and University of Pennsylvania, Philadelphia, PA, United States. [144]Departments of Information Services, Biomedical Informatics, Pediatrics, Cincinnati Children's Hospital Medical Center, University of Cincinnati, Cincinnati, OH, United States. [145]Department of Oncology, ASST Papa Giovanni XXIII, Bergamo, Bergamo, Italy

## Author Contributions

**Conceptualization:** Bertrand Moal, Romain Griffier, Isaac S. Kohane, Alexandre Boyer, Vianney Jouhet.

**Data curation:** Bertrand Moal, Kelly Cho, Romain Griffier, David A. Hanauer, Christian Haverkamp, Yuk-Lam Ho, Meghan R. Hutch, Jeffrey G. Klann, Trang T. Le, Yuan Luo, Adeline Makoudjou, Michele Morris, Danielle L. Mowery, Lav P. Patel, Malarkodi J.

Samayamuthu, Fernando J. Sanz Vidorreta, Emily R. Schriver, Petra Schubert, Shyam Visweswaran, Griffin M. Weber, Zongqi Xia, Alexandre Boyer, Vianney Jouhet.

**Formal analysis:** Bertrand Moal, Antoine Neuraz, Gabriel A. Brat, Paul Avillach, Clara-Lea Bonzel, Tianxi Cai, Kelly Cho, Romain Griffier, Yuk-Lam Ho, Chuan Hong, Ne Hooi Will Loh, Adeline Makoudjou, Michele Morris, Lav P. Patel, Shyam Visweswaran, Xuan Wang, Zongqi Xia, William Yuan, Daniela Zöller, Alexandre Boyer, Vianney Jouhet.

**Investigation:** Bertrand Moal, Arthur Orieux, Sébastien Cossin.

**Methodology:** Bertrand Moal.

**Validation:** Bertrand Moal, Arthur Orieux, Sébastien Cossin.

**Visualization:** Arthur Orieux.

**Writing – original draft:** Bertrand Moal, Thomas Ferté, Antoine Neuraz, Gabriel A. Brat, Paul Avillach, Romain Griffier, David A. Hanauer, Meghan R. Hutch, Trang T. Le, Ne Hooi Will Loh, Yuan Luo, Adeline Makoudjou, Danielle L. Mowery, Karen L. Olson, Guillaume Verdy, Xuan Wang, Griffin M. Weber, Zongqi Xia, William Yuan, Harrison G. Zhang, Isaac S. Kohane, Alexandre Boyer, Vianney Jouhet.

**Writing – review & editing:** Bertrand Moal, Arthur Orieux, Thomas Ferté, Antoine Neuraz, Gabriel A. Brat, Paul Avillach, Sébastien Cossin, Romain Griffier, David A. Hanauer, Meghan R. Hutch, Trang T. Le, Ne Hooi Will Loh, Yuan Luo, Adeline Makoudjou, Danielle L. Mowery, Karen L. Olson, Guillaume Verdy, Xuan Wang, Griffin M. Weber, Zongqi Xia, William Yuan, Harrison G. Zhang, Isaac S. Kohane, Alexandre Boyer, Vianney Jouhet.

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
