## [Decision Letter · Decision Letter 0]

14 Jul 2022

PONE-D-22-09248Acute respiratory distress syndrome after SARS-CoV-2 infection on young adult population: international observational federated study based on electronic health records through the 4CE consortium ARDS after SARS-CoV-2 infection on young adultPLOS ONE

Dear Dr. Moal,

Thank you for submitting your manuscript to PLOS ONE; I sincerely apologise for the unusually delayed review timeframe. Your manuscript has been assessed by one reviewer, whose comments are appended below. After careful consideration, we feel that it has merit but does not fully meet PLOS ONE’s publication criteria as it currently stands. Although the reviewer notes that "The data were of valuable and importan[t]" they nevertheless raise concerns that should be addressed, including the discussion of potential confounders and further information about treatments given to patients in the cohort. Therefore, we invite you to submit a revised version of the manuscript that addresses the points raised during the review process. Please note that we have only been able to secure a single reviewer to assess your manuscript. We are issuing a decision on your manuscript at this point to prevent further delays in the evaluation of your manuscript. Please be aware that the editor who handles your revised manuscript might find it necessary to invite additional reviewers to assess this work once the revised manuscript is submitted. However, we will aim to proceed on the basis of this single review if possible.

We look forward to receiving your revised manuscript.

Kind regards,

Emily Chenette

Editor in Chief

PLOS ONE

Journal Requirements:

2. During your revisions, please confirm whether the wording in the title is correct and compolete, and update it in the manuscript file and online submission information if needed.

There are no competing interests to report.

7. One of the noted authors is a group or consortium, The Consortium for Clinical Characterization of COVID-19 by EHR (4CE) . In addition to naming the author group, please list the individual authors and affiliations within this group in the acknowledgments section of your manuscript. Please also indicate clearly a lead author for this group along with a contact email address.

Reviewers' comments:

Reviewer's Responses to Questions

**Comments to the Author**

1. Is the manuscript technically sound, and do the data support the conclusions?

Reviewer #1: Partly

2. Has the statistical analysis been performed appropriately and rigorously? 

Reviewer #1: Yes

3. Have the authors made all data underlying the findings in their manuscript fully available?

Reviewer #1: No

4. Is the manuscript presented in an intelligible fashion and written in standard English?

Reviewer #1: Yes

5. Review Comments to the Author

Reviewer #1: The authors investigated the risk factors and outcomes of acute respiratory distress syndrome (ARDS) associated with SARS-CoV-2 infection in younger generations trough a federated analysis with large international cohorts. They found the poor outcome of ARDS associated with COVID-19 in younger generations and several diseases (chronic pulmonary disease, diabetes, obesity, paralysis, peptic ulcer) were risk factors for ARDS. The data were of valuable and importance, however, because of federated analysis (coding data) misunderstandings may happen from the results of the present study.

As stated in the Discussion, comorbidities and complications are confusing and misunderstanding. It is unknown that the high mortality rate at day 90 was due to COVID-19 phenotype or comorbidities in the cohort. It is also unclear the management of ARDS in the nature of federated analysis. Were all patient treated in the ICU? The use of respirator, HFNC, and/or ECMO should be demonstrated. Coagulation disorder is he main pathogenesis in severe COVID-19. However, it is impossible to show the relationship between coagulation disorder and severity of COVID-19-ARDS from the coding data.

6. PLOS authors have the option to publish the peer review history of their article (what does this mean?). If published, this will include your full peer review and any attached files.

Reviewer #1: No

---

## [Author Response · Author response to Decision Letter 0]

4 Sep 2022

Response to reviewer

RE: entitled Acute respiratory distress syndrome after SARS-CoV-2 infection on young adult population: international observational study based on electronic health records through the 4CE consortium

Dear Editor and Reviewer,

Thank you very much for your comments and constructive criticisms targeted at improving our manuscript submission. Below you will find our responses to the comments. The manuscript has also been modified accordingly.

Reviewer #1: 

The authors investigated the risk factors and outcomes of acute respiratory distress syndrome (ARDS) associated with SARS-CoV-2 infection in younger generations trough a federated analysis with large international cohorts. They found the poor outcome of ARDS associated with COVID-19 in younger generations and several diseases (chronic pulmonary disease, diabetes, obesity, paralysis, peptic ulcer) were risk factors for ARDS. The data were of valuable and importance, however, because of federated analysis (coding data) misunderstandings may happen from the results of the present study.

As stated in the Discussion, comorbidities and complications are confusing and misunderstanding. It is unknown that the high mortality rate at day 90 was due to COVID-19 phenotype or comorbidities in the cohort.

It is perfectly correct whereas all those patients have been diagnosed with ARDS. We add the following text to the discussion. 

“Mortality at 30 days for ARDS patients of any age with COVID-19 was reported at 39%[4] and corresponds to the mortality for the older ARDS population in our study. The young ARDS population’s mortality at 90 days was smaller, around 16.2% with large variability between HS [11.2; 36.8%]. Importantly, it was not possible to assess the attributable COVID-19 mortality from our data. “

It is also unclear the management of ARDS in the nature of federated analysis. Were all patient treated in the ICU? 

Identification of ICU admissions we not available. It appears that the accuracy of this information in EHR data was poor. It was particularly true at the beginning of the pandemic, where hallways were converted into ad hoc ICUs to support the surge of sick patients, without notification in the chart. This issue has already been discussed in a previous article from the consortium (4). 

This missingness has been added to the limitation.

The use of respirator, HFNC, and/or ECMO should be demonstrated. 

Through the 4CE consortium, the use of respirator, HFNC, and/or ECMO were not available. The collection of treatment data, described by specific codes in EHR, has proven to be too partial and largely heterogeneous between health system (even from the same country) to be collected. 

This element has been added to the limitation.

“Treatment like the use of mechanical ventilation, ECMO or even ICU admission were not collected. The collection of treatment data, described by specific codes in EHR, has proven to be too partial and largely heterogeneous between health systems (even from the same country) to be collected. It also appears that the accuracy of ICU admission in EHR data was poor. It was particularly true at the beginning of the pandemic, where hallways were converted into ad hoc ICUs to support the surge of sick patients, without notification in the chart. This issue has already been discussed in a previous article from the consortium [17].”

Coagulation disorder is the main pathogenesis in severe COVID-19. However, it is impossible to show the relationship between coagulation disorder and severity of COVID-19-ARDS from the coding data.

It is perfectly true, and we agree that EHR/coding data are not the correct tool to highlight this association and it was not the objective of the study. 

Data sharing 

The aggregate data would be shared. They will include for each center, the number and percentage of patients per age groups, per sex, per Elixhauser comorbidities, per complications for all groups.

---

## [Decision Letter · Decision Letter 1]

11 Oct 2022

PONE-D-22-09248R1Acute respiratory distress syndrome after SARS-CoV-2 infection on young adult population: international observational federated study based on electronic health records through the 4CE consortium ARDS after SARS-CoV-2 infection on young adultPLOS ONE

Dear Dr. Moal,

Thank you for submitting your manuscript to PLOS ONE. After careful consideration, we feel that it has merit but does not fully meet PLOS ONE’s publication criteria as it currently stands. Therefore, we invite you to submit a revised version of the manuscript that addresses the points raised during the review process.

Please revise.

We look forward to receiving your revised manuscript.

Kind regards,

Academic Editor

PLOS ONE

Journal Requirements:

Reviewers' comments:

Reviewer's Responses to Questions

**Comments to the Author**

1. If the authors have adequately addressed your comments raised in a previous round of review and you feel that this manuscript is now acceptable for publication, you may indicate that here to bypass the “Comments to the Author” section, enter your conflict of interest statement in the “Confidential to Editor” section, and submit your "Accept" recommendation.

Reviewer #1: (No Response)

Reviewer #2: (No Response)

Reviewer #3: (No Response)

2. Is the manuscript technically sound, and do the data support the conclusions?

Reviewer #1: Yes

Reviewer #2: Yes

Reviewer #3: Yes

3. Has the statistical analysis been performed appropriately and rigorously? 

Reviewer #1: I Don't Know

Reviewer #2: Yes

Reviewer #3: No

4. Have the authors made all data underlying the findings in their manuscript fully available?

Reviewer #1: Yes

Reviewer #2: Yes

Reviewer #3: No

5. Is the manuscript presented in an intelligible fashion and written in standard English?

Reviewer #1: Yes

Reviewer #2: Yes

Reviewer #3: Yes

6. Review Comments to the Author

Reviewer #1: The authors responded well. I have no comment on the revised munuscprist including publication and research ethics.

Reviewer #2: This study by Moal et al identifies risk factors associated with the development of ARDS following COVID-19 in young (18-49 years old) hospitalized adults. Further analysis compares this cohort with older (>49 years old) COVID-ARDS patients. They utilized electronic health records and federated (coded) analysis with multicentre international cohorts. Poor outcome of younger COVID-19-ARDS patients was associated with chronic pulmonary disease, diabetes, obesity, paralysis and peptic ulcers.

The way the data was collected makes it difficult to distinguish between comorbidities and complications. I am unsure why comorbidity data were not restricted to 1 year prior to diagnosis for the entire cohort.

Were the severe COVID-ARDS patients ventilated and could some of these patients have developed ventilator associated lung injury, independently of COVID-19? The definition of severe COVID-19 usually would include some indication of oxygen requirement (either face mask, CPAP or ventilator support). The authors definition of severe COVID-19 could be that they had severe disease along with a COVID-19 infection.

Did patients have critical illnesses (e.g. severe trauma, sepsis, etc) that were the primary cause of ARDS?

It seems that the most important comparator group (severe COVID with no ARDS) were removed from these analyses. This could have introduced biases comparing two extremes of disease (hospitalised mild disease vs severe COVID-ARDS). These analyses could simply be identifying risk factors associated with severe disease and not ARDS specifically.

I wonder whether the authors have split these analyses into smaller age ranges, i.e. 18−25, 26−49, 50−69, 70−79 and 80+ years old, or are the numbers per group too small. I wouldn’t class an individual of 49 years of age to be a young adult (compared to someone 50 years as old).

Was ARDS criteria the same for each site? i.e. as per Berlin criteria.

Reviewer #3: Thank’s to the authors for this interesting piece of work. The difficulty of conducting an international survey was mentioned in this study by peer reviewers and authors. Misunderstandings remain about the methodology used for patient recruitment since there is no consensual health database on all health systems explored. The definition of the keywords searched in the electronic databases had to appear in the methodology.

Also, how did you discriminate hospitalizations whose causes were other than COVID-19?

How do you justify the choice of a poorly frequented hospital in Germany? And what is the benefit of having included this health center when its presence does not influence the results of the study?

7. PLOS authors have the option to publish the peer review history of their article (what does this mean?). If published, this will include your full peer review and any attached files.

Reviewer #1: No

Reviewer #2: No

Reviewer #3: No

---

## [Author Response · Author response to Decision Letter 1]

27 Oct 2022

RE: entitled Acute respiratory distress syndrome after SARS-CoV-2 infection on young adult population: international observational study based on electronic health records through the 4CE consortium

Dear Editor and Reviewer,

Thank you very much for your comments and constructive criticisms targeted at improving our manuscript submission. Below you will find our responses to the comments. The manuscript has also been modified accordingly.

Reviewer #1: The authors responded well. I have no comment on the revised munuscprist including publication and research ethics.

Thank you 

Reviewer #2: This study by Moal et al identifies risk factors associated with the development of ARDS following COVID-19 in young (18-49 years old) hospitalized adults. Further analysis compares this cohort with older (>49 years old) COVID-ARDS patients. They utilized electronic health records and federated (coded) analysis with multicentre international cohorts. Poor outcome of younger COVID-19-ARDS patients was associated with chronic pulmonary disease, diabetes, obesity, paralysis and peptic ulcers.

The way the data was collected makes it difficult to distinguish between comorbidities and complications. I am unsure why comorbidity data were not restricted to 1 year prior to diagnosis for the entire cohort.

Thanks for this remark which highlights a limit of our study. Indeed, the difficulty to distinguish between comorbidities and complications comes from the way data are collected. Comorbidities and complications are identified through ICD codes. In electronic health records, each code is attached to one specific hospitalization visit. For patients with prior hospitalizations, comorbidities are easily identified with the codes attached to those previous hospitalizations. However, for patient without previous hospitalization, the fact that electronic health records do not contain any code, does not mean that patient did not have comorbidities. For example, an obese patient without prior hospitalization would be identified as “obese“ only if we take into account the code associated with the index hospitalization. 

For young ARDS patients, only 31% (312/1001) of them had a previous hospitalization. 

This explains the fact that for our analysis, we use also the code related to the index hospitalization to identify comorbidities. To clarify this limit, we added in the discussion the following paragraph: 

“A limitation of relying on billing codes to identify comorbidities is the challenge of accurately distinguishing comorbidities from complications. In our analysis, comorbidities were considered as those diagnoses from billing codes assigned up to one year before and up to 90 days after the admission. In electronic health records, each code is attached to one specific hospitalization visit. For patients with prior hospitalizations, comorbidities are easily identified with the codes attached to those previous hospitalization. However, for patient without previous hospitalization, the fact that electronic health records do not contains any code, do not mean that patient did not have comorbidities. For example, an obese patient without prior hospitalization would be identified as “obese“ only if we take into account the code associated with the index hospitalization ”.

Were the severe COVID-ARDS patients ventilated and could some of these patients have developed ventilator associated lung injury, independently of COVID-19? The definition of severe COVID-19 usually would include some indication of oxygen requirement (either face mask, CPAP or ventilator support). The authors definition of severe COVID-19 could be that they had severe disease along with a COVID-19 infection. Did patients have critical illnesses (e.g. severe trauma, sepsis, etc) that were the primary cause of ARDS?

As mentioned in the discussion, the treatment like the use of mechanical ventilation, ECMO, or even ICU admission were not collected. Regarding the relation between ARDS and COVID-19 infection, patients included in our analysis had positive reverse transcription PCR tests for SARS-CoV-2 infection 7 days before to 14 days after the date of admission. This inclusion criterion allows us to ensure that included patients had the COVID-19 infection at least at the beginning of the hospitalization. Even if it is not possible to establish a clear temporal relationship or causality between ARDS and COVID-19 with ICD codes, it would be extremely rare that the development of ARDS during the hospitalization of a COVID-19 positive patient had no relation to COVID-19 infection. It is possible that COVID-19 infection was not the primary cause of the ARDS but most likely had an impact on the ARDS development.

To clarify this point, we added in the discussion the following paragraph:

“Regarding the relation between ARDS and COVID-19 infection, patients included in our analysis had positive reverse transcription PCR tests for SARS-CoV-2 infection 7 days before to 14 days after the date of admission. This inclusion criterion allows us to ensure that included patients had the COVID-19 infection at least at the beginning of the hospitalization. Even if it is not possible to establish a clear temporal relationship or causality between ARDS and COVID-19 with ICD codes, it would be extremely rare that the development of ARDS during the hospitalization of a COVID-19 positive patient had no relation to COVID-19 infection. It is possible that COVID-19 infection was not the primary cause of the ARDS but most likely had an impact on the ARDS development.”

It seems that the most important comparator group (severe COVID with no ARDS) were removed from these analyses. This could have introduced biases comparing two extremes of disease (hospitalised mild disease vs severe COVID-ARDS). These analyses could simply be identifying risk factors associated with severe disease and not ARDS specifically.

As noticed, bias of selection could have been introduced because patients with mechanical ventilation, sedatives/anaesthetics or treatment for shock but without an ARDS code (SEVERE_NO_ARDS) were not included to identify comorbidities associated with ARDS.

The choice, to not include them in this comparison, was conducted to eliminate potential miscoded ARDS patients and patients with severe disease or care not related to SARS-CoV-2 infections but with a concomitant infection. 

Those patients could have been included in the ARDS population, but the objective of this study was to focus precisely on ARDS patients, and this grouping would have resulted in a significant measurement bias. Especially because the number of young SEVERE_NO_ARDS patients is greater than the one of ARDS patients. Moreover, for the detection of ARDS patients, a correct sensibility is expected as billing code is related to reimbursement in most countries and ARDS is associated with heavy care. 

Another option would have been to group those patients with the NO_SEVERE patients, but this option would also result in a significant measurement bias. 

In addition, we believe that the descriptive analysis of the SEVERE_NO_ARDS brings credits to this choice (e-table 1). Indeed, the young SEVERE_NO_ARDS population had a higher percentage of women (SEVERE_NO_ARDS: 52%, ARDS: 33%, No SEVERE: 44%). They also had a higher percentage of patients with previous contact with the healthcare system (SEVERE_NO_ARDS: 72%, ARDS: 31%, No SEVERE: 57%). Moreover, 15.1% of the SEVERE_NO_ARDS patients had a billing code associated with pregnancy, 36.1% with long-term drug therapy. These results suggest that the COVID-19 infection was simply concomitant but not the main cause of these hospitalizations.

To clarify this point, we added in the discussion the following paragraph:

To identify comorbidities associated with ARDS following hospitalization with COVID, a comparison was performed considering only non severe patients. Patients with mechanical ventilation, sedatives/anesthetics, or treatment for shock but without ARDS code were not included, which could generate a selection bias. This choice was conducted to eliminate potential miscoded ARDS patients and patients with severe disease or care not related to SARS-CoV-2 infection but with a concomitant infection. Those patients could have been included in the ARDS population, but the objective of this study was to focus precisely on ARDS patients, and this grouping would have resulted in a significant measurement bias. Especially because the number of young SEVERE_NO_ARDS patients is greater than the one of ARDS patients. In addition, we believe that the descriptive analysis of the SEVERE_NO_ARDS brings credit to this choice (e-Table 1). Compared to the other groups, SEVERE_NO_ARDS population had the higher percentage of women (52.2%) and of patients with previous contact with the healthcare system (72%). In addition, 15.1% of those patients had a billing code associated with pregnancy and 36.1% with long-term drug therapy. These results suggest that the COVID-19 infection was simply concomitant but not the main cause of these hospitalizations.

I wonder whether the authors have split these analyses into smaller age ranges, i.e. 18−25, 26−49, 50−69, 70−79 and 80+ years old, or are the numbers per group too small. I wouldn’t class an individual of 49 years of age to be a young adult (compared to someone 50 years as old).

Through the 4CE consortium, the age is not collected as a numeric value but by age groups (18−25, 26−49, 50−69, 70−79, and 80+ years old), as described in the method section. It is an intrinsic limit of our analysis, but this choice has been made to bring more security/de-identification to the data collection process which permits an easier regulatory process for international aggregated data sharing.

To highlight this limit, we have added in the discussion the following sentences:

“More detailed analysis on age’s threshold was not possible because age was intentionally not collected by the 4CE consortium as a continuous variable. This choice was made to ensure greater security/de-identification on the data collection process which allowed for an easier regulatory process for international aggregated data sharing”

Was ARDS criteria the same for each site? i.e. as per Berlin criteria.

The criteria for ARDS detection were the same for each site; ARDS patients were detected with the ICD10 code J80 - Acute respiratory distress syndrome. In each healthcare system the order to code J80 was to meet the Berlin criteria. 

Reviewer #3: Thank’s to the authors for this interesting piece of work. The difficulty of conducting an international survey was mentioned in this study by peer reviewers and authors. Misunderstandings remain about the methodology used for patient recruitment since there is no consensual health database on all health systems explored. The definition of the keywords searched in the electronic databases had to appear in the methodology.

No keywords were searched in the electronic databases. Comorbidities and complications were Identified with billing code. Each health system was using a common classification (ICD10) for those billing codes. The recruitment was based on the visits date and the presence of a positive PCR SARS-CoV-2 test.

Also, how did you discriminate hospitalizations whose causes were other than COVID-19?

The cause of hospitalization was not collected in the analysis. As explained in an answer to the reviewer 2, whereas it is not possible to establish a clear temporal relationship or causality between ARDS and COVID-19 with ICD codes, it would be extremely rare that the development of ARDS during the hospitalization of a COVID-19 positive patient had no relation with COVID-19 infection. It is possible that COVID-19 infection was not the primary cause of the ARDS but most likely had an impact on the ARDS development. For patients without ARDS, COVID-19 could only be a concomitant infection to the hospitalization linked to a different cause, which makes it a good control population

How do you justify the choice of a poorly frequented hospital in Germany? And what is the benefit of having included this health center when its presence does not influence the results of the study?

Thank you very much to notice this typo in the Table 1. The number of Inpatient in the German hospital was not 715 but 71500 patients. By mistake in this table, the numbers finishing by a 0, had lost their 0. The Table 1 has been corrected.

---

## [Decision Letter · Decision Letter 2]

10 Nov 2022

Acute respiratory distress syndrome after SARS-CoV-2 infection on young adult population: international observational federated study based on electronic health records through the 4CE consortium ARDS after SARS-CoV-2 infection on young adult

PONE-D-22-09248R2

Dear Dr. Moal,

We’re pleased to inform you that your manuscript has been judged scientifically suitable for publication and will be formally accepted for publication once it meets all outstanding technical requirements.

Kind regards,

Academic Editor

PLOS ONE

Additional Editor Comments (optional):

Reviewers' comments:

Reviewer's Responses to Questions

**Comments to the Author**

1. If the authors have adequately addressed your comments raised in a previous round of review and you feel that this manuscript is now acceptable for publication, you may indicate that here to bypass the “Comments to the Author” section, enter your conflict of interest statement in the “Confidential to Editor” section, and submit your "Accept" recommendation.

Reviewer #1: All comments have been addressed

Reviewer #2: All comments have been addressed

2. Is the manuscript technically sound, and do the data support the conclusions?

Reviewer #1: Yes

Reviewer #2: Yes

3. Has the statistical analysis been performed appropriately and rigorously? 

Reviewer #1: I Don't Know

Reviewer #2: Yes

4. Have the authors made all data underlying the findings in their manuscript fully available?

Reviewer #1: Yes

Reviewer #2: No

5. Is the manuscript presented in an intelligible fashion and written in standard English?

Reviewer #1: Yes

Reviewer #2: Yes

6. Review Comments to the Author

Reviewer #1: The authors responded to reviewers` comments and revised well. The manuscript is clearly presented and will be of interest to readers of PLOS ONE.

Reviewer #2: (No Response)

7. PLOS authors have the option to publish the peer review history of their article (what does this mean?). If published, this will include your full peer review and any attached files.

Reviewer #1: No

Reviewer #2: No

---

## [Editor Report · Acceptance letter]

13 Dec 2022

PONE-D-22-09248R2 

Acute respiratory distress syndrome after SARS-CoV-2 infection on young adult population: international observational federated study based on electronic health records through the 4CE consortium 

Dear Dr. Moal:

I'm pleased to inform you that your manuscript has been deemed suitable for publication in PLOS ONE. Congratulations! Your manuscript is now with our production department. 

Kind regards, 

on behalf of

Dr. Robert Jeenchen Chen 

Academic Editor

PLOS ONE